# The Yeast Permease Agp2 Senses Cycloheximide and Undergoes Degradation That Requires the Small Protein Brp1-Cellular Fate of Agp2 in Response to Cycloheximide

**DOI:** 10.3390/ijms24086975

**Published:** 2023-04-10

**Authors:** Ashima Mohanty, Abdallah Alhaj Sulaiman, Balasubramanian Moovarkumudalvan, Reem Ali, Mustapha Aouida, Dindial Ramotar

**Affiliations:** 1Division of Genomics and Precision Medicine, College of Health and Life Sciences, Hamad Bin Khalifa University, Education City, Qatar Foundation, Doha P.O. Box 34110, Qatar; 2Division of Biological and Biomedical Sciences, College of Health and Life Sciences, Hamad Bin Khalifa University, Education City, Qatar Foundation, Doha P.O. Box 34110, Qatar

**Keywords:** uptake transporter, sensor, drugs, protein synthesis inhibitor, cycloheximide, ubiquitinylation, protein degradation, a small protein, yeast

## Abstract

The *Saccharomyces cerevisiae* Agp2 is a plasma membrane protein initially reported to be an uptake transporter for L-carnitine. Agp2 was later rediscovered, together with three additional proteins, Sky1, Ptk2, and Brp1, to be involved in the uptake of the polyamine analogue bleomycin-A5, an anticancer drug. Mutants lacking either Agp2, Sky1, Ptk2, or Brp1 are extremely resistant to polyamines and bleomycin-A5, suggesting that these four proteins act in the same transport pathway. We previously demonstrated that pretreating cells with the protein synthesis inhibitor cycloheximide (CHX) blocked the uptake of fluorescently labelled bleomycin (F-BLM), raising the possibility that CHX could either compete for F-BLM uptake or alter the transport function of Agp2. Herein, we showed that the *agp2Δ* mutant displayed striking resistance to CHX as compared to the parent, suggesting that Agp2 is required to mediate the physiological effect of CHX. We examined the fate of Agp2 as a GFP tag protein in response to CHX and observed that the drug triggered the disappearance of Agp2 in a concentration- and time-dependent manner. Immunoprecipitation analysis revealed that Agp2-GFP exists in higher molecular weight forms that were ubiquitinylated, which rapidly disappeared within 10 min of treatment with CHX. CHX did not trigger any significant loss of Agp2-GFP in the absence of the Brp1 protein; however, the role of Brp1 in this process remains elusive. We propose that Agp2 is degraded upon sensing CHX to downregulate further uptake of the drug and discuss the potential function of Brp1 in the degradation process.

## 1. Introduction

The yeast *Saccharomyces cerevisiae* Agp2 protein was initially shown to mediate the uptake of L-carnitine into yeast cells [1]. L-carnitine is required to carry acetyl-CoA generated by fatty acid β-oxidation via the peroxisomal-mitochondrial carnitine acetyltransferase (Cat2) trafficking to permit complete oxidation in the mitochondria by the Krebs cycle [1]. Further studies identified Agp2 as a low-affinity, non-specific amino acid permease that operates in poor nutrient conditions [2]. Agp2 was subsequently classified as a regulated transporter that belongs to the family of amino acid permeases consisting of several members, including Put4, Alp1, Lyp1, Can1, and Gap1 [2,3,4]. However, its redundant function and low affinity and specificity with several active amino acid permeases such as Gap1 suggest that Agp2 could have other function(s). In a search for haploid yeast gene-deletion mutants with increased resistance to an analogue of the anticancer drug bleomycin, bleomycin-A5 (BLM-A5), a spermidine-conjugated bleomycin species that damage the DNA, we rediscovered the *AGP2* gene, as well as three additional genes *PTK2, SKY1,* and *BRP1* [5]. Mutants devoid of the proteins Agp2, Ptk2, Sky1, or Brp1 were extremely resistant to bleomycin [5]. Moreover, these mutants, in particular the *agp2Δ* mutant, lost the ability to accumulate fluorescently labelled bleomycin (F-BLM) into the cells consistent with the resistance to the drug [5]. Interestingly, a short pre-treatment with the protein synthesis inhibitor CHX did not interfere with the initial uptake of F-BLM that occurred during the first 60 min, but the subsequent uptake beyond this time was completely inhibited [6]. We interpreted the finding to suggest that the transport of F-BLM occurred in two phases, an early phase and a late phase, and proposed that the latter phase of uptake may depend upon, for example, new protein synthesis of the Agp2 transporter. However, subsequent findings, as highlighted below, challenged this proposition and raised new possibilities.

The *agp2Δ* mutants are also extremely resistant to the toxic effects of spermine and spermidine that coincided with the sharp reduction in the initial velocity of labelled spermine and spermidine uptake using similar approaches that identified Agp2 as a carnitine permease [1,7]. As such, Agp2 has been befitted the first eukaryotic protein identified as a polyamine permease in yeast [7]. However, Uemura et al., 2007 independently reported the isolation of two new high-affinity permeases, Sam3 and Dur3, that contribute to the major pool of polyamine uptake in yeast [8]. This raised the possibility that Agp2 may act as a regulator that positively influences the activity level of Sam3 and Dur3.

*S. cerevisiae* has several regulators acting as non-transporting transceptors (sensors), such as the plasma membrane protein Ssy1, belonging to the same family of amino acid permease family as Agp2, that cannot function as a transporter [9,10,11]. Ssy1 senses the level of amino acids and triggers the expression of several amino acid permeases that include *AGP1*, *BAP2, BAP3, DIP5,* and *TAT1*. It acts through a mechanism that involves the proteolytic activation of two cytosolic transcription factors, Stp1 and Stp2, which translocate to the nucleus to activate the expression of the amino acid permease genes [9,12,13]. The growing list of plasma membrane permeases acting as sensors and the ability of Agp2 to recognize and mediate the uptake of L-carnitine, bleomycin, and polyamines prompted a reassessment of the role of this transporter [1,6,7,14]. A variety of gene expression approaches were undertaken and the analysis revealed that cells lacking Agp2 lost the ability to maintain the proper expression of nearly 172 genes when compared to the parent [15]. Some of these genes under the control of Agp2 encode membrane transporters such as *SAM3*, *DUR3*, *HNM1*, *TPO2*, and *HXT3*, raising the possibility that Agp2 controls the uptake of diverse substrates [8,16,17,18,19,20]. Thus, the reduced uptake of F-BLM in the presence of CHX could be explained if CHX acts to compete with bleomycin uptake or can inhibit Agp2 function.

In this study, we set out to investigate whether Agp2 is involved in mediating the uptake of CHX. We showed that the *agp2Δ* mutant, and not the *ptk2Δ*, *sky1Δ*, or *brp1Δ* mutants, displayed resistance to CHX as compared to the parent, suggesting that CHX physiological effect is mediated via Agp2. The double mutant lacking Sam3 and Dur3, two of the transporters controlled by Agp2, also exhibited resistance to CHX, but not to the same extent as the *agp2Δ* mutant raising the possibility that Agp2 is a major contributor to the cellular response to CHX. As such, we monitored the fate of Agp2 as a GFP tag protein in response to CHX and observed that the drug triggered the disappearance of Agp2 in a concentration- and time-dependent manner. Immunoprecipitation analysis revealed that Agp2-GFP exists in higher molecular weight forms that were ubiquitinylated, which disappeared within 10 min of treatment with CHX, suggesting that Agp2 may undergo CHX-induced proteolysis. We further found that CHX did not significantly trigger the degradation of Agp2-GFP in the absence of the Brp1 protein. We interpret these data to suggest that Agp2 is degraded upon sensing the toxic compound CHX to downregulate further uptake of the drug. We discuss the potential role of Brp1 in the degradation of Agp2 and highlight the possibility that Agp2 could serve as a master regulator in detecting and controlling the entry of toxic compounds into the cell.

## 2. Results

### 2.1. Protein Modelling Studies Reveal Several Structural Features of Agp2

The yeast Agp2 protein sequence from the Uniprot database consisted of 596 amino acids with a calculated molecular mass of 67.2 kDa [21]. Visual representation of the transmembrane topology of Agp2 using Protter encompassed 12 transmembrane domains with the N- and C-terminals facing the cytoplasm (Figure 1) [22].

In addition, the prediction indicated that Agp2 possessed ten loops, five facing the outward side and five facing the inward side, with respect to the plasma membrane (Figure 1). Agp2 possessed four lysine residues (lysine20, lysine22, lysine24, and lysine595) that are ubiquitinylated and a serine residue (serine45) that is phosphorylated (see the *Saccharomyces* Genome Database, https://www.yeastgenome.org, accessed on 16 September 2021). A similar pattern of ubiquitinylated residues was reported for the high-affinity tryptophan amino acid permease, Tat2 [23].

The 3D structural model of Agp2 was predicted using the I-TASSER web server [24]. The extracellular, intracellular, and side views of the modelled Agp2 with the N-and C-terminal being inside of the cell are consistent with the Protter analysis (Figure 2).

The I-TASSER predicted model of Agp2 with the highest confidence score (C-Score: −1.96) was further validated using the Ramachandran plot generated by the PROCHECK program [25] inbuilt within PDBsum webserver (Appendix A) [26].

The Ramachandran plot showed that 67.3% of the amino acid residues were in the favourable region, 24.6% were in the additionally allowed region, and 5.7% were in the generously allowed region. Overall, 97.6% of the amino acid residues were in the allowed region indicating a good stereochemical quality of the predicted Agp2 model (Appendix A). The model was next analyzed for secondary structural details using the PDBSum web server [26]. As shown in Appendix A, the predicted Agp2 protein model encompassed 23 α helices, including 12 transmembrane helices and 2 β sheets forming an alpha-helical barrel structure, which predominantly constitutes alpha helices.

### 2.2. Protein-Ligand Docking Reveals That CHX Binds to Agp2

We next assessed whether the tertiary structure of Agp2 could accommodate the binding of CHX using protein–ligand docking studies. The Agp2-CHX docking study revealed that CHX can bind to Agp2 with a docking score of −6.48 kcal/mol (Figure 3a,b).

The analysis suggests that CHX is a substrate that can be recognized by Agp2. The Agp2–CHX interactions were visualized using LigPlot+ (Figure 3c) [27]. Hydrogen bonds were observed to be formed by residues Pro368 (2.75 Å), Tyr369 (2.53 Å), Ile371 (3.15 Å), and Ala372 (3.06 Å). These residues were inherent in the extracellular loop between transmembrane (TM) helices TM-7 and TM-8 (Figure 1 and Appendix A). Apart from the hydrogen bond forming residues, the residues Tyr259, Val345, Cys346, Ala360, Ser367, Val370, and Asn374, inherent in the extracellular loops between TM5-TM6 and TM7-TM8 were also observed to interact with CHX. Thus, suggesting that these flexible extracellular loops are involved in the recognition of CHX. Furthermore, the domain architecture of the Agp2 protein was analyzed using the InterPro domain database (https://www.ebi.ac.uk/interpro/, accessed on 27 January 2023) [28]. The interacting residues of Agp2 were located within the defined region assigned to the domain 92-563 for the amino acid permease SLC12A (Appendix A).

### 2.3. Spot Test and Growth Curve Analyses Reveal That agp2Δ Mutants Are Resistant to CHX

If CHX is a substrate recognized by Agp2, then in the absence of Agp2, the resulting *agp2Δ* mutant cells would be expected to show resistance to the drug, as seen for bleomycin and polyamines [5,7]. To test this, the WT and the *agp2Δ* mutant were grown overnight in liquid YPD media and then the next day, the cells were adjusted to an OD_600nm_ of ~0.6 before being serially diluted to 1:10, 1:100, 1:1000, and 1:10,000 and spotted onto plates without and with increasing concentrations of CHX (from 0 to 0.075 µg/mL). The *agp2Δ* mutant is known to grow slower, but this phenotype did not affect its responses to drugs [6]. As shown in Figure 4A, the *agp2Δ* mutant grew onto plates that contained the highest concentration (0.075 µg/mL) of CHX, while the WT showed minimal growth.

In a previous study, the *agp2Δ* mutant was identified together with three other mutants, *ptk2Δ*, *sky1Δ*, and *brp1Δ,* showing resistance to the anti-cancer drug bleomycin and polyamines, raising the possibility that these proteins may all function in the same pathway [5,15]. As such, we checked if these mutants, *ptk2Δ*, *sky1Δ*, and *brp1Δ,* would also be resistant to CHX. To our surprise, none of these three mutants, *ptk2Δ*, *sky1Δ*, and *brp1Δ,* showed resistance to CHX, raising the possibility that Agp2 performs a unique role distinct from Ptk2, Sky1, and Brp1 towards the resistance to CHX (Figure 4A). Of note, a double mutant *sam3Δ dur3Δ* lacking two polyamine uptake transporters, Sam3 and Dur3, whose expressions are controlled by Agp2, also showed resistant to CHX (Figure 4A) [15].

In an independent analysis, the WT and the five mutants (*agp2Δ, ptk2Δ*, *sky1Δ*, *brp1Δ*, and the double mutant *sam3Δ dur3Δ*) were sub-cultured at a low optical density of ~0.2 in the absence and presence of CHX (0.1 µg/mL) in liquid YPD media, and the growth of the strains was monitored over time using a plate reader equipped with an orbital shaker. While all the strains grew and reached an OD_600nm_ of ~1.8 by 16 h in the absence of CHX, only the *agp2Δ* mutant and the double mutant *sam3Δ dur3Δ* exhibited growth in the presence of CHX (Figure 4C,D). These results suggest that cells lacking Agp2 are indeed resistant to CHX and that Agp2 may be involved in mediating the uptake of CHX into the cells perhaps via the uptake transporters Sam3 and Dur3. Based on the resistance level to CHX, Agp2 appears to exert a more prominent role.

### 2.4. agp2Δ Mutant Displays WT Resistance to Rapamycin

We examined if the *agp2Δ* mutant would display resistance to the unrelated drug rapamycin. Rapamycin is an immunosuppressant used in organ transplants and for treating various cancers. It acts by inhibiting the TOR kinase pathway and mimicking starvation conditions leading to cell cycle arrest in the G1 phase [29,30,31]. We performed spot test and growth curve analyses with rapamycin under similar conditions as for CHX, and found that the *agp2Δ* mutant showed normal WT resistance to rapamycin, excluding the possibility that Agp2 is involved in the recognition of a broad array of drugs (Appendix A).

### 2.5. CHX Triggers Agp2 Disappearance in a Time-and Concentration-Dependent Manner

Since Agp2 is required to protect cells from the toxic effects of CHX, we checked whether this transporter would undergo changes when the cells are exposed to CHX. No commercial antibodies are available to monitor Agp2 levels, therefore we used our previously published single-copy plasmid pAGP2-GFP designed to express Agp2 from its native promoter as a GFP fusion protein for detection by immunoblot analysis [5,6,32,33]. The plasmid pAGP2-GFP expressed a functional Agp2-GFP fusion protein as it rescued the uptake of fluorescently labelled bleomycin into the *agp2Δ* mutant [5]. pAGP2-GFP was introduced into the WT strain and the cells were treated with increasing concentrations of CHX (0 to 10 μM) for 30 min before the samples were used for the preparation of crude plasma membrane [34]. Since Agp2 was previously shown to be present on the plasma membrane, we used this fraction for monitoring Agp2-GFP levels by immunoblot analysis probed with anti-GFP antibodies. For these experiments, we did not boil the crude extract to analyze the Agp2-GFP by immunoblot as plasma membrane proteins are known to aggregate upon boiling and generate broad diffuse bands [34]. In addition, GFP is known to form aggregates upon boiling [34]. We, therefore, decided to heat the crude plasma membrane at 65 °C for 10 min in the SDS loading buffer before processing it onto SDS-PAGE. In the absence of CHX treatment, we observed in several experiments a diffused band representing the partially unfolded protein that ran in the range of 65 to 75 kDa, instead of the estimated size (88 kDa) of Agp2-GFP (Figure 5A, lane 1; and see Appendix A for the full blot, and an independent experiment Appendix A).

In addition, the mobility of Agp2-GFP varied depending on the quality of the gels, i.e., commercial *versus* laboratory-made gels (Figure 5A, lane 1; and the Appendix A). Upon treatment with increasing concentrations of CHX, there was a significant decrease in the level of Agp2-GFP at 10 μM for 30 min as compared to the untreated cells, suggesting that CHX induces the disappearance of Agp2 (Figure 5A, lane 3 and see Appendix A for the full blot; Appendix A for an independent experiment; 5B, Ponceau staining of the immunoblot to monitor for protein loading; and 5C, quantification of the Agp2-GFP protein). No further decrease of Agp2-GFP was observed if the cells were treated with 20 μM CHX for 30 min, suggesting that there is a residual fraction of Agp2-GFP in the crude plasma membrane that resists the loss triggered by CHX (see Appendix A).

In separate experiments, we examined whether the disappearance of Agp2-GFP would be influenced by the time of treatment with CHX. For this experiment, cells were treated with a fixed concentration of CHX (10 μM) and monitored for the loss of Agp2-GFP at 0, 15 and 30 min using immunoblot analysis of the extracted crude plasma membrane fraction. Agp2-GFP began to disappear by 15 min of CHX treatment and by 30 min less than 20% of the protein remained (Figure 5D, lane 2 and 3 vs. 1; 5E showed the Ponceau staining of the immunoblot to monitor for protein loading; and 5F showed the quantification of the Agp2-GFP protein). The data support the notion that CHX induces the disappearance of Agp2, which may occur via protein degradation.

### 2.6. CHX-Induced Disappearance of Agp2-GFP Is Defective in the brp1Δ and ptk2Δ Mutants

We checked whether the genes *BRP1*, *PTK2,* and *SKY1* involved in the bleomycin- and polyamine-resistant pathway would play a role in CHX-induced Agp2-GFP disappearance [5,7]. Both Ptk2 and Sky1 are kinases that regulate high-affinity polyamine uptake into the cell, while the role of Brp1 has not yet been defined, although there is some partial evidence that it may be involved in regulating the expression of the *PMA1* gene encoding the proton pump Pma1 [5,35]. We examined the level of expression of Agp2-GFP when the single copy plasmid pAGP2-GFP was introduced into the WT and the gene-deletion mutants *brp1Δ* and *ptk2Δ*. The plasma membrane fraction was isolated from the WT and the mutants and monitored for Agp2-GFP level by immunoblot analysis. We observed that the basal level of Agp2-GFP was nearly the same in both mutants as compared to the WT (Figure 6A, lanes 4 and 7 vs. lane 1, which contained slightly more extract; see Figure 6B for the immunoblot stained with Ponceau).

Upon treatment with CHX, the Agp2-GFP was reduced in the WT as expected, but not in the *brp1Δ* or *ptk2Δ* mutant (Figure 6A, lane 3 vs. lanes 6 and 9, respectively; and quantified in Figure 6C; see also Appendix A for the consistent effect of the *brp1Δ* mutant).

This finding indicates that Brp1 and Ptk2 contribute directly or indirectly to facilitate the disappearance of Agp2-GFP in response to the CHX. Since Ptk2 is known to have a defined role, that is, to phosphorylate Pma1 to activate its function [17], we focused primarily on investigating whether Brp1 has a role in triggering Agp2 disappearance.

### 2.7. Immunoprecipitated Agp2-GFP Contains High Molecular Weight Species That Are Ubiquitinylated and Disappear upon CHX Treatment in WT, Not in the brp1Δ Mutant

Several permeases including the high-affinity tryptophan permease Tat2 are removed from the plasma membrane by ubiquitinylation, which triggers their internalization and transport to the vacuole for degradation [23]. We investigated whether CHX would trigger the appearance of high molecular weight forms of Agp2-GFP. In this experiment, exponentially growing WT and the *brp1Δ* mutant expressing Agp2-GFP were untreated and treated with the indicated dose of CHX and the total extracts were subjected to immunoprecipitation with magnetic GFP-trap beads coupled with anti-GFP antibodies followed by immunoblot analysis [36] (see Section 4). The immunoprecipitation experiment was conducted to enrich for any possible weakly expressed forms of Agp2-GFP. The data revealed that the primary form of Agp2-GFP pull down by the beads existed as a diffused band of ~75 kDa in size in the absence of CHX treatment (Figure 7A, lane 1).

At least 80% of the immunoprecipitated Agp2-GFP was not recovered upon CHX treatment for 15 min (Figure 7A, lane 4). In parallel, we examined the form of Agp2-GFP immunoprecipitated from the *brp1Δ*/pAGP2-GFP strain. As in the WT, the same diffuse band was seen in the *brp1Δ* mutant expressing Agp2-GFP (Figure 7A, lane 5 vs. 1), however, there was no loss of the protein in response to the CHX treatment (Figure 7A, lane 8 vs. 5), consistent with the data observed from the plasma membrane fraction. This finding supports the notion that Brp1 is required to mediate the disappearance of Agp2-GFP in response to CHX exposure.

It is noteworthy that under this growth and treatment condition, there were no major high molecular weight forms of Agp2-GFP detected in the exponentially growing WT or the *brp1Δ* mutant, suggesting that if CHX induces the appearance of high molecular weight species, these forms may be too weak to be detected by the anti-GFP antibodies.

Stress-induced disappearance of proteins often involves ubiquitinylation of the proteins followed by proteasomal degradation [37]. We examined whether Agp2-GFP could be ubiquitinylated and which may play a role in its disappearance upon CHX treatment. As such, we conducted an independent immunoprecipitation experiment with the above total extracts using the magnetic GFP trap beads to check if the immunoprecipitated fraction of Agp2-GFP would contain ubiquitinylated species. As shown in Figure 7B, several high molecular weight species were detected by the anti-ubiquitin antibodies from the immunoprecipitate derived from the WT (lanes 1 to 4). These high molecular weight species were degraded in the WT following 15 min of CHX treatment (lane 4; in this blot, most of the lane 1 sample was lost). In contrast, none of the immunoprecipitated high molecular weight ubiquitinylated species detected in the *brp1Δ*/pAGP2-GFP strain appeared to be significantly diminished in response to CHX treatment (Figure 7B lanes 5 to 8). These findings suggest that the ubiquitinylated species of Agp2-GFP are likely marked for degradation and the process is stimulated when the cells are challenged with the toxic compound CHX. The data also reveal that Brp1 plays a role in the degradation process, but not in the ubiquitinylation of Agp2-GFP.

### 2.8. Saturated Cultures Contain Significantly High Levels of Ubiquitinylated Agp2-GFP Species, Which Are Susceptible to CHX-Induced Loss in the WT and Not in the brp1Δ Mutant

Since saturated cultures are generally more resistant to drugs, we checked whether this growth condition would influence the expression level of Agp2-GFP [38]. The WT and *brp1Δ* mutant expressing Agp2-GFP were grown for 48 h and total protein extracts were subjected to immunoprecipitation using the anti-GFP trap magnetic beads. When the same amounts of extracts were immunoprecipitated and analyzed by immunoblot, several prominent high molecular weight polypeptides of Agp2-GFP were detected in the absence of CHX treatment (Figure 8A, lanes 1 and 4, respectively, for the WT and the *brp1Δ* mutant; Figure 8B showing the input Agp2-GFP; Figure 8C, lanes 1 and 4, respectively, for the WT and the *brp1Δ* mutant).

Interestingly, these high molecular weight species rapidly disappeared in the WT upon treatment with CHX, but not in the case of the *brp1Δ* mutant (Figure 8A,C lanes 2 and 3 vs lanes 5 and 6, respectively, for the WT and the *brp1Δ* mutant). The data indicate that high molecular weight Agp2-GFP species accumulate in the saturated cultures and these forms appear to be highly susceptible to CHX-induced loss in a manner dependent upon the Brp1 protein.

### 2.9. Expression of Brp1 in the brp1Δ Mutant Re-Instates the Degradation of Agp2-GFP

To confirm that Brp1 is indeed involved in mediating Agp2-GFP disappearance, we designed a plasmid using the single-copy vector YCplac22 to express the entire *BRP1* gene from its promoter and in frame with the MYC tag sequence. The resulting plasmid pBRP1-MYC was sequenced and confirmed to contain the entire BRP1-MYC-tagged gene (see Appendix A). This plasmid pBRP1-MYC and the empty vector were introduced into the *brp1Δ* mutant expressing Agp2-GFP. Total extracts prepared from the strain containing the plasmid pBRP1-MYC did not express any detectable Brp1-MYC protein. We assumed that the Brp1-MYC protein was either very unstable in the total extract despite the presence of protease inhibitors or that its expression was extremely low and cannot be detected by the anti-Myc antibody.

Treatment of the *brp1Δ* mutant cells carrying the plasmid pAGP2-GFP and the empty vector showed no significant reduction of the Agp2-GFP protein upon exposure to CHX (Figure 9, lanes 1 to 4; see similar results in Figure 6, lanes 4 to 6 and Figure 7A, lanes 5 to 8).

In contrast, when the *brp1Δ* mutant carried both the pAGP2-GFP and the pBRP1-MYC plasmids, the loss of the Agp2-GFP protein was re-instated following treatment of the cells with CHX (Figure 9, lanes 5 to 8). The data reveal that although the Brp1 protein level cannot be detected, there is enough expressed to mediate the disappearance of Agp2 in response to CHX treatment.

### 2.10. Intracellular Distribution of Agp2-GFP Is Dependent upon Brp1

We examined the intracellular localization of Agp2-GFP in the WT and the *brp1Δ* mutant to determine whether its distribution would be affected by CHX using a confocal microscope. Agp2-GFP displayed a broad distribution within the WT and *brp1Δ* mutant (Figure 10, panels A and D, respectively).

Interestingly, DAPI staining for the nuclear DNA showed that some of the Agp2-GFP protein was localized to the nucleus in the WT and, to a lesser extent, in the *brp1Δ* mutant, as shown by the merged image (Figure 10 image A vs. D). Upon treatment with CHX (10 µM for 10 min), Agp2-GFP showed drastic redistribution, with most of the protein appearing to be in the cytosol in the WT (Figure 10, image B), and the effect seemed less severe in the *brp1Δ* mutant (Figure 10, image E). Importantly, there was a sharp reduction (>80%) in the amount of Agp2-GFP in the WT after 30 min of treatment (Figure 10, image C, note image taken at 60× magnification). In contrast, there appeared to be no significant loss of the Agp2-GFP protein in the *brp1Δ* mutant in response to the same treatment (Figure 10, image F), consistent with the above findings (Figure 6). Altogether, the data support the notion that Agp2 senses CHX, thereby initiating its degradation by ubiquitinylation in a process requiring Brp1.

## 3. Discussion

This work was undertaken to revisit an earlier observation whereby the protein synthesis inhibitor CHX was shown to block the uptake of fluorescently labelled bleomycin into yeast cells, and we interpreted the finding to suggest that a component of the uptake pathway required new protein synthesis [6]. Herein, we provide in silico evidence showing that CHX can dock onto Agp2 with a very low energy requirement, indicating that it may compete for bleomycin binding onto the transporter, thus explaining the reduced uptake of fluorescently labelled bleomycin into the cells in the presence of CHX. Furthermore, the observation that CHX treatment induced the disappearance of Agp2, as seen with its tagged form, Agp2-GFP, may provide an alternative explanation for the reduced level of fluorescently labelled bleomycin uptake in the cells, as opposed to directly competing for the bleomycin uptake. We believe that the disappearance of Agp2 in response to CHX treatment is not due to inhibition of protein synthesis, as other mutants tested in this study such as *brp1Δ* or *ptk2Δ* that displayed parental sensitivity to CHX do not show a reduction of the Agp2-GFP. Thus, the disappearance of Agp2-GFP induced by CHX is likely a consequence of protein degradation. This is supported by a large-scale screen for K63 ubiquitinylation in yeast that identified many ubiquitinylated proteins, including Agp2, which is ubiquitinylated at four amino acid residues, lysine 20, lysine 22, lysine 24 and lysine 595, which carries the branch modification K63 [39,40]. While these ubiquitinylated residues may serve to maintain the function of Agp2, these modified sites can also serve to initiate polyubiquitinylation to trigger its degradation [40]. This is evident by the high molecular weight ubiquitinylated species of Agp2 that were rapidly lost by CHX treatment (see Figure 8C). In addition to Agp2, several other transporters and sensors have been found to contain K63 modification and are removed from the plasma membrane by ubiquitinylation [9,41]. These include the general amino acid permease Gap1, the proline permease Put4, the ammonium permease Mep2, regulated by the availability of nitrogen sources, as well as many other plasma membrane permeases, such as the constitutively expressed high-affinity tryptophan permease Tat2 [23]. The removal of the transporters and sensors from the plasma membrane triggers the internalization of the transporters and ultimately to the vacuole for degradation [40,42,43]. Thus, it seems plausible that the disappearance of Agp2 in response to CHX is a result of its degradation. We do not know if this implies branch ubiquitinylation of one or all four ubiquitinylated lysines of Agp2, as well as its phosphorylation site on Ser45. 

The most significant finding from our study is the observation that Brp1 is required for the degradation of Agp2 in response to CHX. We showed that deletion of the *BRP1* gene prevented the degradation of Agp2 and re-introduction of the *BRP1* gene into the *brp1Δ* mutant re-instated the degradation of Agp2. The exact role of Brp1 in this process remains to be investigated. It should be noted that the *BRP1* gene is embedded within the promoter of the *PMA1* gene encoding the voltage gradient pump and deletion of *BRP1* caused the downregulation of the *PMA1* gene. However, the absence of Brp1 does not affect the expression of Agp2 even from the plasmid-driven expression of Agp2-GFP using the *AGP2* natural promoter. Thus, it is unlikely that Brp1 downregulates the gene expression level of Agp2 to account for the decreased level of the protein in response to CHX. The evidence so far indicates that Brp1 does not play a role in the ubiquitinylation of Agp2. The immunoprecipitated Agp2-GFP was found to be ubiquitinylated in the presence or absence of Brp1 (Figure 7 and Figure 8).

So what is the role of Brp1? One possibility is that Brp1 may be involved in recruiting Agp2 to be degraded by the proteasomal complex. As such, we might anticipate finding Brp1 in association with Agp2 following CHX treatment. Alternatively, Brp1 may perform a function to direct Agp2 to other intracellular locations such as the vacuoles for degradation. This is a likely possibility as we have observed that Agp2-GFP appears to have a wide cellular distribution and a fraction of the protein is present in the nucleus of WT cells using confocal microscopy. We suggest that Brp1 may be involved in exporting Agp2 from the nucleus to the cytosol for its degradation. The redistribution of Agp2-GFP to the cytosol could be affected in the *brp1Δ* mutant. While Brp1 is required for the degradation of Agp2 in response to CHX, it is important to note that Ptk2 also plays a role in the degradation of Agp2. The Ptk2 kinase could activate the degradation process by phosphorylating Agp2. Whether this involves the known phosphorylated Ser45 of Agp2 remains to be investigated.

Previously, there was no concrete evidence indicating whether Agp2 functions as (i) a sensor that activates downstream targets as in the case of Ssy1 or Gap1, (ii) a transporter only, or (iii) both as a sensor and a transporter. Our present data support the notion that Agp2 could act as a sensor for CHX and not a transporter. Earlier, we showed that Agp2 regulates the expression of several genes of the high-affinity polyamine uptake pathway and these genes include *SAM3* and *DUR3* encoding the two high-affinity polyamine uptake transporters, *SKY1* and *PTK2* encoding kinases that regulate the high-affinity polyamine uptake process, as well as several other transporter genes such as *HNM1* encoding the choline transporter [15]. These transporters can transport other substrates and are considered to have broad substrate specificity. Since the *sam3Δ dur3Δ* double mutant, and not the single mutant, displayed resistance to CHX, it would appear that both Sam3 and Dur3 are involved in the uptake of CHX. The fact that the *sam3Δ dur3Δ* double mutant exhibited CHX resistance in the presence of Agp2 strongly suggests that Agp2 is unlikely to act as a direct transporter of CHX, otherwise it would sensitize the double mutant to the drug. As such, we propose that the major role of Agp2 is to act as a sensor. If, indeed, Agp2 acts as a sensor, it is not clear how it executes this function. One possibility is that Agp2 sends a signal or enters the nucleus to maintain/activate gene expressions such as *SAM3*, *DUR3*, *HNM1*, and *SKY1* [13]. The reduced resistance of the *sam3Δ dur3Δ* double mutant to CHX, as compared to the *agp2Δ* mutant, could be explained if the choline transporter regulated by Agp2 is also a transporter of CHX. Thus, deleting the *HNM1* gene in the *sam3Δ dur3Δ* double mutant to create the triple mutant may confer a similar level of CHX resistance as the *agp2Δ* mutant strain.

In conclusion, this study investigates the mechanism by which the membrane transporter protein Agp2 responds to the cytotoxic effects of CHX. We provide evidence that Agp2 acts as a sensor for CHX that triggers the degradation of Agp2 via ubiquitinylation and requires the small protein Brp1 whose function is under investigation. As such, the degradation of Agp2 would turn off a further influx of CHX into the cell, perhaps by downregulating the function of Dur3 and Sam3. We propose that Agp2 may serve as a master regulator to sense several toxic molecules and undergo ubiquitin-dependent degradation, thereby preventing the accumulation of these deleterious compounds in the cell. As such, cells lacking Agp2 are expected to be resistant to a range of toxic compounds.

## 4. Materials and Methods

### 4.1. Strains, Media, Transformation and Reagents—The S. cerevisiae Strains Used in the Present Study Are Listed in Table 1

Yeast cells were grown at 30 °C in either Yeast Peptone Dextrose (YPD, FORMEDIUM CCM0105) YPD 1% (*w/v*) or minimal synthetic (SD: 0.65% yeast nitrogen base without amino acids, 2% dextrose, 0.17% dropout mix) medium used for transformation [44,45,46]. All strains used were obtained from non-essential haploid mutant resources from this laboratory, except for the strains prepared for this study. All chemical reagents including CHX were purchased from Sigma, St Louis, MA, USA.

**Table 1 ijms-24-06975-t001:** Yeast strains used in this study.

Strains	Genotypes	Sources
BY4741 Wild type	*MAT* *α his3* *Δ leu2* *Δ met15* *Δ ura3* *Δ*	This laboratory collection
BY4741(*sky1Δ*)	Isogenic to BY4741, except *sky1Δ::KAN*	This laboratory collection
BY4741(*brp1Δ*)	Isogenic to BY4741, except *brp1Δ::KAN*	This laboratory collection
BY4741(*ptk2Δ*)	Isogenic to BY4741, except *ptk2Δ::KAN*	This laboratory collection
BY4741(*agp2Δ*)	Isogenic to BY4741, except *agp2Δ::KAN*	This laboratory collection
BY4741 Wild type	carrying the single copy plasmid pAGP2-GFP on *URA3* selection.	This study
BY4741(*agp2Δ*)	carrying the single copy plasmid pAGP2-GFP on *URA3* selection.	This study
BY4741 (*brp1Δ*)	carrying the single copy plasmid pAGP2-GFP on *URA3* selection.	This study
BY4741 (*ptk2Δ*)	carrying the single copy plasmid pAGP2-GFP on *URA3* selection.	This study
BY4741 (*sam3Δ dur3Δ*)	Isogenic to BY4741, except *sam3Δ::HIS3 dur3Δ::KAN*	This laboratory collection
BY4741 (*brp1Δ*)/pAGP2-GFP + vector.	carrying the single copy plasmid pAGP2-GFP on *URA3* selection.	This study
BY4741 (*brp1Δ*)/pAGP2-GFP + pBRP1-MYC.	carrying the single copy plasmids pAGP2-GFP *on URA3* selection and pBRP-MYC on *LEU2* selection	This study

### 4.2. Protein Modelling and Docking Studies

The yeast Agp2 protein sequence was retrieved from the Uniprot database (Uniprot ID: P38090; https://www.uniprot.org/, accessed on 16 September 2021) [21] and the subsequent structural modelling was done using I-TASSER web server (https://zhanggroup.org/I-TASSER/, accessed on 16 September 2021) [24]. The predicted model of the yeast Agp2 with the highest confidence score (C-Score) was further validated using PROCHECK [25] within the PDBsum web server (http://www.ebi.ac.uk/thornton-srv/databases/pdbsum/, accessed on 16 September 2021) [26] and subsequently used for protein–ligand docking studies. The 3D ligand coordinates of cycloheximide (CHX) were downloaded from the PubChem database (PubChem CID: 6197; https://pubchem.ncbi.nlm.nih.gov/, accessed on 17 September 2021) [47]. Both the yeast Agp2 and the ligand were prepared using AutoDock Tools by removing water molecules, protonation and addition of Gasteiger charges before to docking [48]. Protein–ligand docking studies were performed using the MTiAutoDock web service in-built into the MTiOpenScreen web server (https://bioserv.rpbs.univ-paris-diderot.fr/services/MTiOpenScreen/, accessed on 17 September 2021) [49]. PyMOL Molecular Graphics System (Version 2.3, Schrödinger, LLC, New York, NY, USA), UCSF Chimera (Version 1.15, University of California, San Francisco, CA, USA) [50] and LigPlot+ (Version 2.2.4, written by Roman Laskowski, European Molecular Biology Laboratory, European Bioinformatics Institute, Cambridge, UK) [27] software tools were used for molecular visualization and analysis. The transmembrane topology of Agp2 was visualized using the Protter web server (https://wlab.ethz.ch/protter/start/, accessed on 27 January 2023) [22].

### 4.3. Spot Test Analysis of Cell Growth

Standard spot tests were performed as previously described [51]. For the growth curve, the cells were cultured overnight in YPD media, then the OD_600nm_ was adjusted to 0.2 with fresh YPD media in a 96-well plate. Drugs were added to the cells at the indicated concentrations and their growth was monitored for 24 h in a TECAN plate reader with shaking. The temperature was set to 30 °C and the OD was taken every hour. The result was plotted as a graph of OD_600nm_ against the time.

### 4.4. Construction of Plasmid pBRP1-MYC

Primers YCpLac111-BRP1-F2

5′CTGCAGGTCGACTCTAGAGGATCCCTGAATGTGTGTATAAAAGAGAGAAAAAATGG-3′

and YCpLac111-BRP1-R2

5′GTCCTCTTCAGAAATGAGCTTTTGCTCCGTCACCATCGCTGACGGGGAAAAAACAAAACAAGCTTCCTG-3′

were designed to amplify the *BRP1* gene from yeast genomic DNA using PCR (with SmaI and EcoRI restriction sites). The amplified product was digested using SmaI and EcoRI and inserted into the YCpLac111 single-copy plasmid by cloning [45,52] to produce the plasmid pBRP1-MYC. The YCplac111 plasmid lacked a promoter due to which it could not express MYC. The empty vector YCpLac111 and pBRP1-MYC plasmid were introduced separately into the yeast *brp1Δ* mutant.

### 4.5. CHX Treatment

The indicated strains were cultured overnight in 20 mL YPD media at 30 °C. The following day, cells were adjusted to OD_600nm_ of 30, and then treated with 10 μM of CHX for the indicated time points. The drug treatment was stopped by washing the cells with sterile water. The total cells were collected by centrifugation. The plasma membrane was then extracted from the total yeast cells as mentioned below.

### 4.6. Plasma Membrane and Total Cell Extraction

Membrane fractions were prepared as described by [34]. Briefly, yeast cells were cultured overnight at 30 °C with shaking, and log-phase cells were pelleted and re-suspended in yeast suspension buffer (YSB) (50 mM Tris–HCl (pH 7.6), 5 mM EDTA, 10% glycerol, 1× complete protease inhibitor cocktail tablet) along with sterile glass beads (0.5 mm diameter, BioSpec Cat. No. 11079105). Yeast cells were lysed using a bead mill homogenizer (BeadMill 4, FisherScientific, Loughborough, UK) at 5 m/s for 5 s, repeated 10 times with cooling on ice in between. The unbroken cells and glass beads were spun down by brief centrifugation at 3000 rpm for 1 min in an Eppendorf centrifuge and the supernatant containing total cell extracts (aliquots saved as total cell extract) was further centrifuged at 4 °C for 1 h at 15,000 rpm to fractionate crude plasma membrane as a pellet. The pellet was re-suspended in 50 μL YSB. The crude plasma membrane protein fraction was quantified before processing for Western Blot or Immunoprecipitation assays.

### 4.7. Western Blotting

The protein extracts were mixed with 1X SDS loading buffer (4X stock SDS loading buffer recipe: 4 g SDS, 2.5 mL Glycerol, 3.13 mL Stacking buffer (pH 6.8), 1.25 mL β-mercaptoethanol, 10 mg bromophenol blue and made up to 10 mL with sterile water) and sterile water to make the final volume to 20 μL in separate vials and heated at 65 °C for 10 min. The samples were loaded onto a 10–12% SDS-polyacrylamide gel, followed by electrophoresis (PAGE). The gel was transferred onto a nitrocellulose membrane using a wet transfer method. The membranes were blocked with blocking buffer 5% non-fat milk in 1X TBS-Tween (10 mM Tris–HCl pH 7.5, 15 mM NaCl, 1.2 mM EDTA and 0.1% Tween 20) for 1 h. After blocking, the membrane was probed with Primary Anti-GFP antibody at 1:5000 dilution in blocking buffer overnight at 4 °C. The next day, the membranes were washed for 10 min 3 times with 1X TBST. The membranes were then incubated with HRP-Affinity pure (H + L) anti-mouse secondary antibody 1:5000 (Jackson Immunoresearch, Cambridge House, St.Thomas’ Place, UK) for 1 h at room temperature, followed by washing with 1X TBST for 15 min, 3 times. The membranes were incubated with the Thermofischer Western blot ECL solution as per the manual and were visualized using Pierce™ ECL Western Blotting Substrate (Thermofisher Scientific).

### 4.8. Immunoprecipitation Assay

Total cell extracts of yeast cells untreated or treated with CHX were incubated in ChromoTek GFP-Trap Agarose beads (Proteintech) for 2.5 h at 4 °C to pull-down *AGP2-GFP* tagged-associated proteins. After incubation, the beads coupled to proteins were washed with washing buffer (10 mM Tris/Cl pH 7.5, 0.5 mM EDTA, 150 mM NaCl and 0.05% NonidetTM P40 Substitute) 3 times and collected on a magnetic stand. The proteins were eluted in 2X SDS loading dye and heated at 65 °C for 10 min. Proteins were processed on 12% SDS gel and the membranes were probed with an anti-GFP antibody or Anti-Ubiquitin antibody.

### 4.9. Confocal Imaging

Yeast strains were grown overnight and treated with CHX for the indicated time points, followed by washing with sterile water. Cells were resuspended in 300 μL of 4% Formaldehyde for 15 min at RT for fixation. The cells were collected by centrifugation, then resuspended in 500 μL KPO4/Sorbitol solution with 0.5 μL DAPI, and incubated at RT for 30 min. The cells were washed with 500 μL KPO4/Sorbitol Solution 3 times, for 10 min each, and resuspended in 300 μL of wash buffer. A sample of 2 μL of these cells was pipetted onto a glass slide along with 2 μL of glycerol. Slides were covered with coverslips and sealed with nail polish. These glass slides were visualized using A1R confocal microscope and the analysis was performed using ImageJ software.

## Figures and Tables

**Figure 1 ijms-24-06975-f001:**
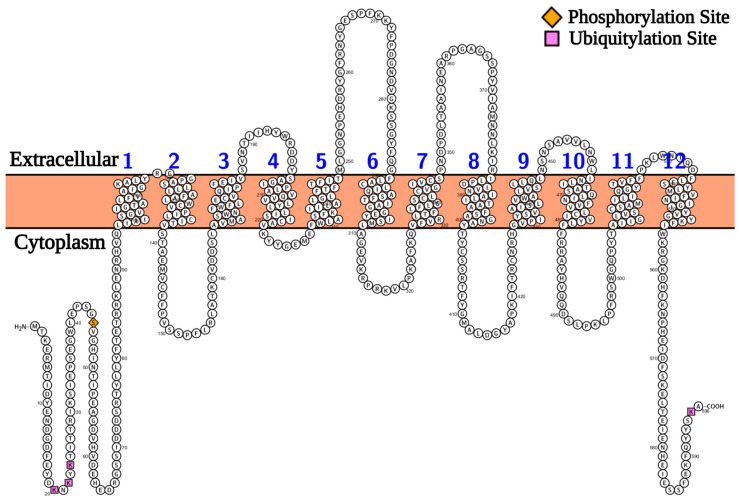
Protter representation of Agp2 transmembrane topology comprising 12 transmembrane helices. The ubiquitylation sites (Lysine 20, 22, 24, and 595) and the phosphorylation site (Serine 45) are indicated as purple square and orange diamond, respectively.

**Figure 2 ijms-24-06975-f002:**
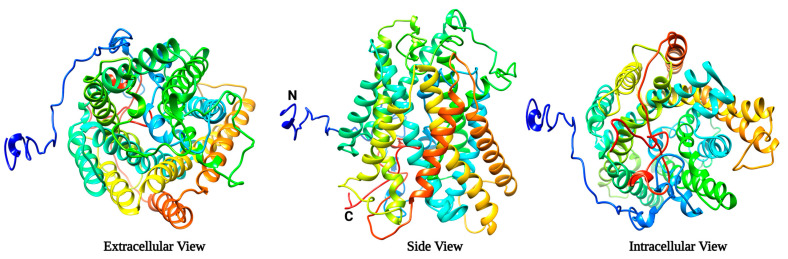
I-TASSER predicted model of Agp2 from different views. Extracellular is the top view from the outside of the lipid bilayer. The side view is from the front looking across the lipid bilayer, N and C denote N- and C-termini, respectively. The intracellular view is from the inside of the cell.

**Figure 3 ijms-24-06975-f003:**
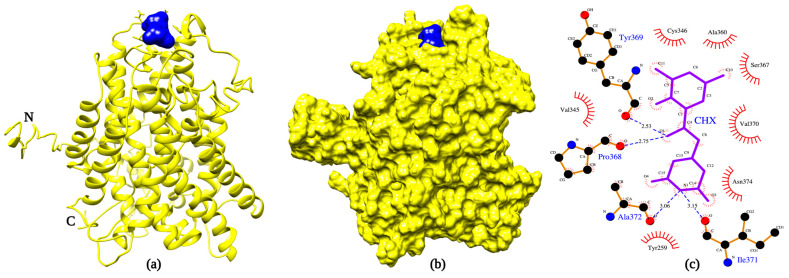
Docked complex structure representing Agp2 bound with CHX. (**a**,**b**) show the ribbon and surface representation of Agp2 in yellow and CHX in blue. (**c**) Ligplot interaction map of Agp2-CHX complex.

**Figure 4 ijms-24-06975-f004:**
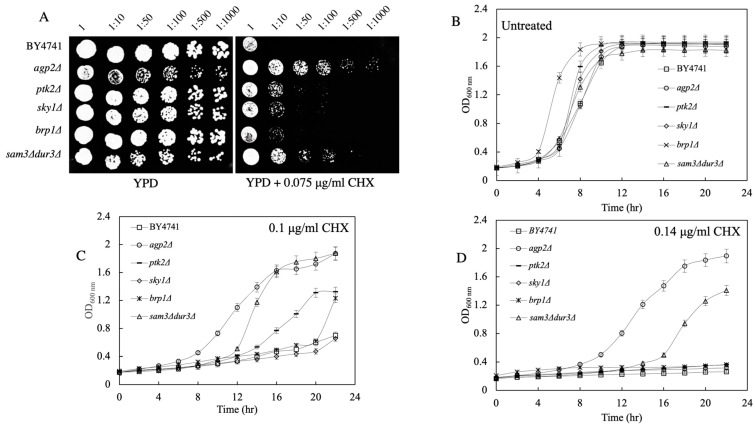
Survival analysis of the indicated strains showing that the *agp2Δ* mutant is resistant to the toxic effects of CHX. (**A**) Spot test analysis. Briefly, overnight cultures were serially diluted and spotted onto solid YPD plates without and with CHX. Plates were photographed after 72 h of incubation at 30 °C. (**B**–**D**) Overnight cultures were adjusted to OD_600nm_ of ~0.2 in fresh YPD liquid media without and with CHX and the growth of the cells was monitored automatically using a plate reader equipped with an orbital shaker. Panel B shows growth of the cells untreated. Panel C and D show growth of the cells in the presence of CHX, 0.1 and 0.14 µg/mL, respectively.

**Figure 5 ijms-24-06975-f005:**
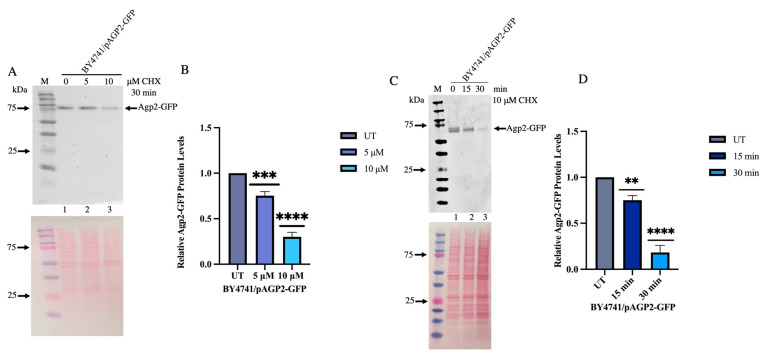
Immunoblot showing Agp2-GFP is disappearing upon exposure to CHX. (**A**) Immunoblot analysis of cells treated with increasing concentrations of CHX. Exponentially growing cells from the wild type (WT) (strain BY4741) expressing Agp2-GFP were treated with 0 to 10 μM CHX for 30 min, followed by crude plasma membrane preparation (see “Section 4”) and analysis by immunoblot probed with anti-GFP antibodies. Ponceau staining was performed to monitor for protein loading. (**B**) Quantification of the disappearance of Agp2-GFP relative to the untreated sample (lane 1). (**C**) Immunoblot analysis of cells treated with a fixed concentration of CHX at different times. The samples were processed as in panel (**A**), showing Ponceau staining below and (**D**) quantification of Agp2 levels. M, prestained protein markers in kDa. Results are representative of three independent experiments. Error bars indicate S.E. The data are representative of three biological replicates and analyzed by 2-way Anova Test. ** Is equivalent to *p*-value < 0.01, *** Is equivalent to *p*-value < 0.001, **** Is equivalent to *p*-value < 0.0001.

**Figure 6 ijms-24-06975-f006:**
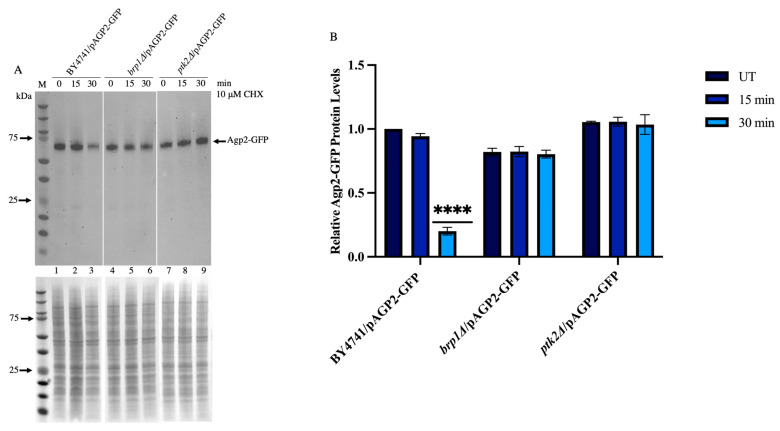
Comparison of Agp2-GFP levels between the WT (BY4741) and the *brp1Δ* and *ptk2Δ* mutants upon treatment with the indicated doses of CHX. (**A**) The crude plasma membrane fraction was prepared as in Figure 5A following treatment with 10 μM CHX for 0, 15 and 30 min and analyzed by immunoblot. Ponceau staining of the immunoblot to monitor for protein loading. (**B**) Quantification of the relative level of Agp2-GFP under the various treatment conditions. M, prestained protein markers in kDa. Error bars indicate S.E. The data are representative of three biological replicates and analyzed by 2-way Anova Test. **** Is equivalent to *p*-value < 0.0001.

**Figure 7 ijms-24-06975-f007:**
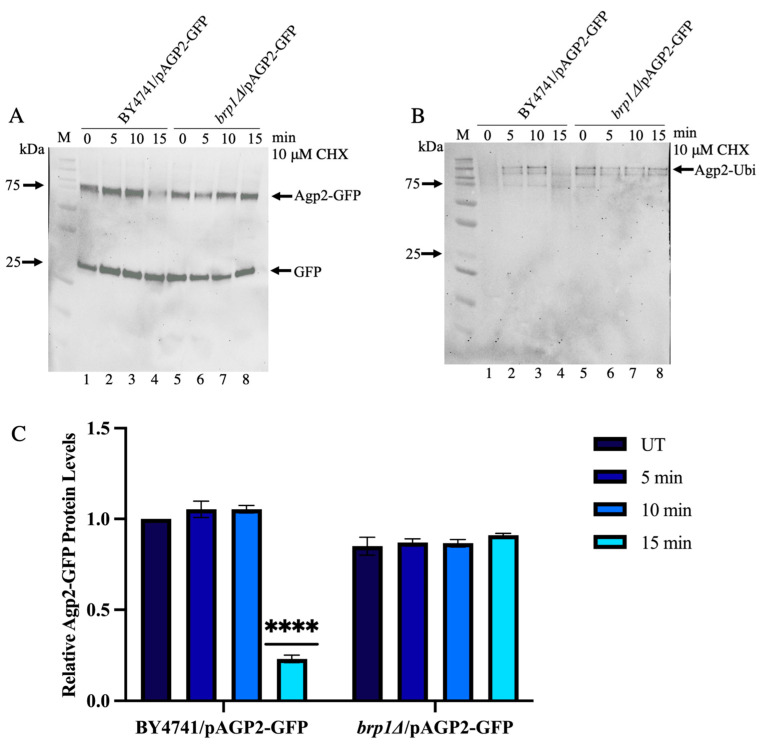
Analysis of immunoprecipitated Agp2-GFP from exponentially growing cultures of the WT and the *brp1Δ* mutant. (**A**) Immunoprecipitation of Agp2-GFP showing its disappearance in the WT and not in the *brp1Δ* mutant following CHX treatment. Briefly, total extracts were prepared from exponentially growing cells treated with CHX, subjected to pull-down assay using anti-GFP magnetic beads and probed by immunoblot with anti-GFP antibodies. (**B**) The same amount of the pull-down samples from panel (**A**) were analyzed by immunoblot probed with anti-ubiquitin antibodies. (**C**) Quantification of the relative level of the immunoprecipitated Agp2-GFP under the indicated conditions. M, prestained protein markers in kDa. Error bars indicate S.E. The data are representative of three biological replicates and analyzed by 2-way Anova Test. **** Is equivalent to *p*-value < 0.0001.

**Figure 8 ijms-24-06975-f008:**
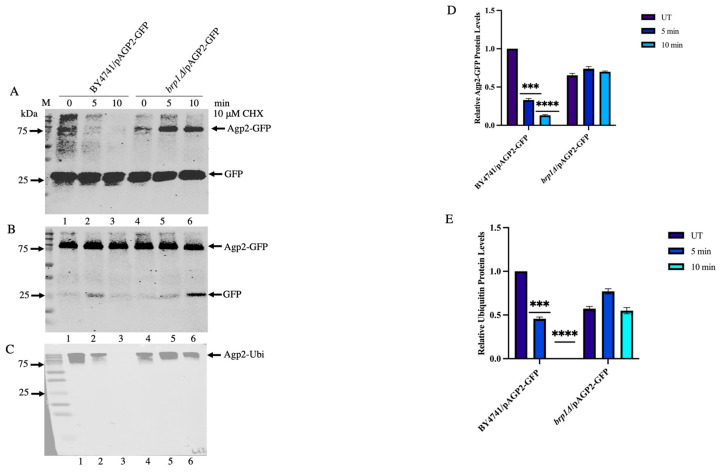
Analysis of immunoprecipitated Agp2-GFP from saturated cultures of the WT and the *brp1Δ* mutant. (**A**) Immunoprecipitation of Agp2-GFP showing its disappearance in the WT and not in the *brp1Δ* mutant following CHX treatment. Briefly, total extracts were prepared from saturated cultures treated with CHX, subjected to pull-down assay using anti-GFP magnetic beads and probed by immunoblot with anti-GFP antibodies. (**B**) Total cell extracts were used to monitor the input level of Agp2-GFP. (**C**) The same amount of the pull-down samples from panel (**A**) were analyzed by immunoblot probed with anti-ubiquitin antibodies. (**D**,**E**) Quantification of the relative level of the immunoprecipitated Agp2-GFP under the indicated conditions from panels (**A**,**C**), respectively. M, prestained protein markers in kDa. Error bars indicate S.E. The data are representative of three biological replicates and analyzed by 2-way Anova Test. *** Is equivalent to *p*-value < 0.001, **** Is equivalent to *p*-value < 0.0001.

**Figure 9 ijms-24-06975-f009:**
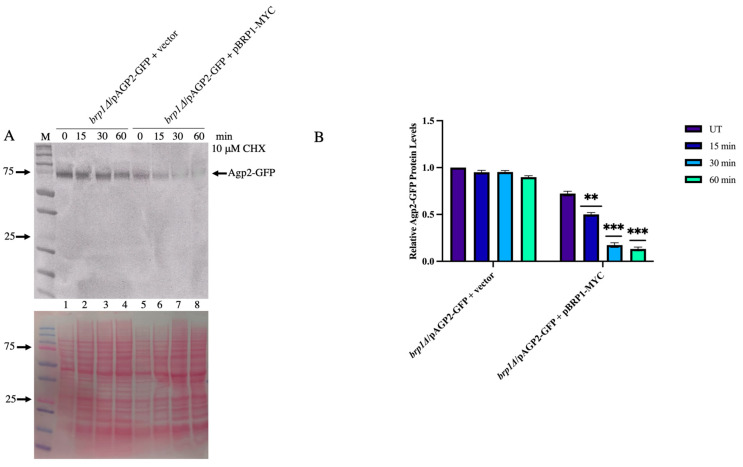
Re-introduction of the *BRP1* gene into the *brp1Δ* mutant reinstates the disappearance of Agp2-GFP in response to CHX. (**A**) Exponentially growing cells of the *brp1Δ* mutant expressing AGP2-GFP and carrying either the empty vector or the BRP1-MYC plasmid were treated with 10 μM CHX followed by plasma membrane extraction and immunoblot analysis. Ponceau staining was performed to monitor for protein loading. (**B**) Quantification of the relative level of Agp2-GFP in the *brp1Δ* mutant carrying either the empty vector or the BRP1-MYC plasmid under the indicated treatment conditions. M, prestained protein markers in kDa. Error bars indicate S.E. The data are representative of three biological replicates and analyzed by 2-way Anova Test. ** Is equivalent to *p*-value < 0.01, *** Is equivalent to *p*-value < 0.001.

**Figure 10 ijms-24-06975-f010:**
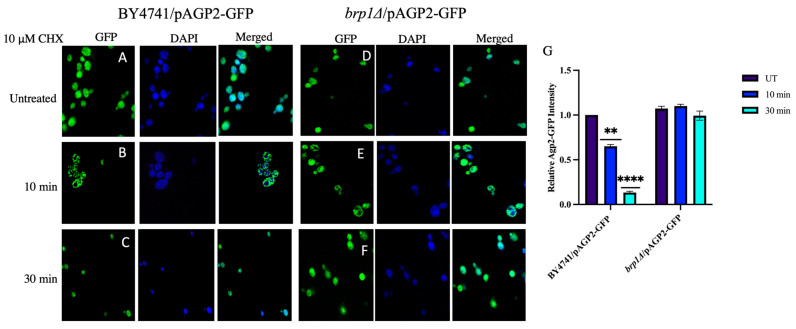
Confocal microscopy showing the distribution of Agp2-GFP in the WT and *brp1Δ* mutant in response to CHX. Briefly, exponentially growing cells were treated with CHX, washed, fixed, and imaged using confocal microscopy. (**A**,**B**,**D**–**F**) Cells were imaged at 100× magnification and panel (**C**) at 60× magnification. (**G**) Quantification of Agp2-GFP in the indicated strains following CHX treatment. Error bars indicate S.E. The data are representative of three biological replicates and analyzed by 2-way Anova Test. ** Is equivalent to *p*-value < 0.01, **** Is equivalent to *p*-value < 0.0001.

## Data Availability

The datasets used and/or analyzed during the current study are included in the manuscript.

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
