# Peer review of "The Yeast Permease Agp2 Senses Cycloheximide and Undergoes Degradation That Requires the Small Protein Brp1-Cellular Fate of Agp2 in Response to Cycloheximide"

_ijms, 2023, doi:10.3390/ijms24086975_

Round 1

Reviewer 1 Report

Summary

The manuscript submitted by Mohanty and co-workers describes the investigation of the role of the yeast permease, Agp2, in cycloheximide (CHX) uptake. The authors show that mutants lacking Agp2 show resistance to CHX, and that ubiquitinylated forms of Agp2 are degraded over time upon treatment with CHX, but only in the presence of the Brp1 protein. The authors propose that Agp2 acts as a sensor for CHX that is degraded upon treatment with CHX to downregulate uptake of this toxic compound.

Comments

1. Did the authors consider any transcriptomic analysis upon treatment with CHX? This could give more information as to whether the effects seen are due to up/down-regulation of transporters/proteases.

2. I noticed that different sized fonts are used throughout the text (lines 20-24 for example), consistent font size should be used.

3. In each figure, a red figure indicator is present, this should be removed as the figure legend already denotes figure number.

4. Supplementary figures appear to be in the main manuscript? These should be moved to a supplementary file.

Author Response

Summary

The manuscript submitted by Mohanty and co-workers describes the investigation of the role of the yeast permease, Agp2, in cycloheximide (CHX) uptake. The authors show that mutants lacking Agp2 show resistance to CHX, and that ubiquitinylated forms of Agp2 are degraded over time upon treatment with CHX, but only in the presence of the Brp1 protein. The authors propose that Agp2 acts as a sensor for CHX that is degraded upon treatment with CHX to downregulate uptake of this toxic compound.

Comments

Comment 1: Did the authors consider any transcriptomic analysis upon treatment with CHX? This could give more information as to whether the effects seen are due to up/down-regulation of transporters/proteases.

Response 1: Thank you.  We have conducted RNA seq analysis for the wild type and the agp2 mutant only under untreated Agp2 condition (treated condition under way).  We have also done mass spectrometry analysis for both strains under untreated and treatment with CHX.  The analysis revealed that there are various changes at the transcriptomic levels between the wild type and the agp2 mutant. One of the key findings, is that the pleiotropic drug efflux pump Pdr5 is the upregulated in the agp2 mutant and explains the resistance of the agp2 mutant to CHX.  Pdr5 has been previously shown to efflux CHX. The RNA seq and mass spectrometry data are part of an ongoing work for another manuscript.

Comment 2. I noticed that different sized fonts are used throughout the text (lines 20-24 for example), consistent font size should be used.

Response 2: We have made the changes.

Comment 3. In each figure, a red figure indicator is present, this should be removed as the figure legend already denotes figure number.

Response 3: We have made the changes.

Comment 4. Supplementary figures appear to be in the main manuscript? These should be moved to a supplementary file.

Response 4: We have removed the supplementary data from the main body of the manuscript and included these as a separate file.  

Reviewer 2 Report

The authors have investigated whether Agp2 is involved in mediating the 84 uptake of CHX. They have designed several experiments. But they have never explained why this is important. The significance or goal of the work is undefined. There is no conclusion written. I do not think this work will be of interest to readers. Therefore, I can not recommend this paper for publication in a high impact journal IJMS.

Author Response

The authors have investigated whether Agp2 is involved in mediating the 84 uptake of CHX. They have designed several experiments.

Comments: But they have never explained why this is important. The significance or goal of the work is undefined.

Response: Point well taken and we have now indicated at the significance of the work.  As such, we add to the end of the introduction the following  “...and highlight the possibility that Agp2 could serve as a master regulator in detecting and controlling the entry of toxic compounds into the cell”.

Comment: There is no conclusion written.

Response: We thank the reviewer and now added a conclusion at the end of the discussion as follows: “In conclusion, this study investigates the mechanism by which the membrane transporter protein Agp2 responds to the cytotoxic effects of CHX.  We provide evidence that Agp2 acts as a sensor for CHX that triggers the degradation of Agp2 via ubiquitinylation and requires the small protein Brp1 whose function is under investigation.  As such, the degradation of Agp2 would turn off a further influx of CHX into the cell perhaps by downregulating the function of Dur3 and Sam3. Agp2 may serve as a master regulator to sense several toxic molecules and undergo ubiquitin-dependent degradation thereby preventing the accumulation of these deleterious compounds in the cell.  As such, cells lacking Agp2 are expected to be resistant to a range of toxic compounds”.  

Reviewer 3 Report

Author here describe a plasma membrane associated transmembrane protein, Agp2 that is involved in interacting with drug cycloheximide to undergo degradation requiring Brp1. Overall, manuscript provides several evidences to test the Agp2-CHX interaction and the Agp2 degradation. Following are some of the minor point that need to be address. 

1. In Fig 4 A., why is agp2 mutant growing better on YPD medium containing CHX as compared to YPD medium lacking the CHX?

2. The plate assay were done for 2 ng RAP concentrations, but liquid culture with 100ng, could you please explain the reason?

3. Were the unequal protein loading from Figures S5 and other taken into consideration while quantifying the blots?

Author Response

Author here describe a plasma membrane associated transmembrane protein, Agp2 that is involved in interacting with drug cycloheximide to undergo degradation requiring Brp1. Overall, manuscript provides several evidences to test the Agp2-CHX interaction and the Agp2 degradation. Following are some of the minor point that need to be address. 

Comment 1: In Fig 4 A., why is agp2 mutant growing better on YPD medium containing CHX as compared to YPD medium lacking the CHX?

Response 1: The image for YPD + CHX plate was taken at 72 hrs, while the plate for YPD was taken at 48 hrs. The agp2 mutant in fact grows slightly slower, but reaches the same final optical density as the wild type (Fig. 4B).  It should be noted that this slight grown delay does not interfere with agp2 ability to confer drug resistance. 

  1. The plate assay were done for 2 ng RAP concentrations, but liquid culture with 100ng, could you please explain the reason?

Response 2: The plate assay was a chronic treatment for 72 hrs, therefore, low concentration of the drug is sufficient to see a difference in growth.  For the liquid culture, the treatment was for 24 hours and higher concentration of drug is required to see if there is a difference in the growth of the mutants and the WT.

  1. Were the unequal protein loading from Figures S5 and other taken into consideration while quantifying the blots?

Response 3: Thank you and yes indeed.  The quantification was done for the Agp2 band against a specific band in their respective ponceau stain.